# Estimation of change in pleural pressure in assisted and unassisted spontaneous breathing pediatric patients using fluctuation of central venous pressure: A preliminary study

**Nao Okuda**[1,2], **Miyako Kyogoku**[2], **Yu Inata**[2], **Kanako Isaka**[2], **Kazue Moon**[2], **Takeshi Hatachi**[2], **Yoshiyuki Shimizu**[2], **Muneyuki Takeuchi**[2] *

**1** Center for Infectious Disease, Nara Medical University Hospital, Kashihara-shi, Nara, Japan, **2** Department of Intensive Care Medicine, Osaka Women's and Children's Hospital, Izumi-shi, Osaka, Japan

* mutake1017@gmail.com

## Abstract

### Background

It is important to evaluate the size of respiratory effort to prevent patient self-inflicted lung injury and ventilator-induced diaphragmatic dysfunction. Esophageal pressure (Pes) measurement is the gold standard for estimating respiratory effort, but it is complicated by technical issues. We previously reported that a change in pleural pressure (ΔPpl) could be estimated without measuring Pes using change in CVP (ΔCVP) that has been adjusted with a simple correction among mechanically ventilated, paralyzed pediatric patients. This study aimed to determine whether our method can be used to estimate ΔPpl in assisted and unassisted spontaneous breathing patients during mechanical ventilation.

### Methods

The study included hemodynamically stable children (aged <18 years) who were mechanically ventilated, had spontaneous breathing, and had a central venous catheter and esophageal balloon catheter in place. We measured the change in Pes (ΔPes), ΔCVP, and ΔPpl that was calculated using a corrected ΔCVP (cΔCVP-derived ΔPpl) under three pressure support levels (10, 5, and 0 cmH$_2$O). The cΔCVP-derived ΔPpl value was calculated as follows: cΔCVP-derived ΔPpl = k × ΔCVP, where k was the ratio of the change in airway pressure (ΔPaw) to the ΔCVP during airway occlusion test.

### Results

Of the 14 patients enrolled in the study, 6 were excluded because correct positioning of the esophageal balloon could not be confirmed, leaving eight patients for analysis (mean age, 4.8 months). Three variables that reflected ΔPpl (ΔPes, ΔCVP, and cΔCVP-derived ΔPpl) were measured and yielded the following results: -6.7 ± 4.8, − -2.6 ± 1.4, and − -7.3 ± 4.5

**Data Availability Statement:** All relevant data are within the manuscript and its Supporting Information files.

**Funding:** This work was supported by JSPS KAKENHI Grant Number JP18K08904.

**Competing interests:** The authors have declared that no competing interests exist.

cmH2O, respectively. The repeated measures correlation between cΔCVP-derived ΔPpl and ΔPes showed that cΔCVP-derived ΔPpl had good correlation with ΔPes (r = 0.84, p< 0.0001).

## Conclusions

ΔPpl can be estimated reasonably accurately by ΔCVP using our method in assisted and unassisted spontaneous breathing children during mechanical ventilation.

## Introduction

Mechanical ventilation is a life-saving measure in patients with respiratory failure. However, excessive unloading of the respiratory muscles by mechanical ventilation causes ventilator-induced diaphragmatic dysfunction (VIDD), which in turn prolongs the need for mechanical ventilation [1]. Similarly, vigorous respiratory efforts and insufficient respiratory muscle unloading by mechanical ventilation can cause patient self-inflicted lung injury (P-SILI) and damage the respiratory muscles, which also prolongs mechanical ventilation [2–4]. Therefore, it is important to maintain optimal respiratory effort to protect both the lung and the diaphragm during mechanical ventilation [5, 6].

Respiratory effort can be estimated by measuring the pleural pressure (Ppl) [5, 6]. In clinical practice, esophageal pressure (Pes), determined using an esophageal balloon catheter, is used as a surrogate for Ppl [7]. However, the measurement of Pes is complicated by technical issues, including those related to the correct positioning of the esophageal catheter, interpretation of absolute Pes values, and balloon volume [8, 9]. As a potential surrogate for the detecting the change in Ppl (ΔPpl) or strong inspiratory efforts, the change in central venous pressure (ΔCVP) has been repeatedly examined [10–18]. However, inconsistent results in previous papers have shown ΔCVP to be both an underestimation and an overestimation of ΔPpl [10, 12, 14–18]. Accordingly, ΔCVP has not been generally accepted as a surrogate for ΔPpl.

We previously reported that ΔPpl could be estimated with reasonable accuracy using the ΔCVP when it is adjusted with a simple correction method in mechanically ventilated, paralyzed pediatric patients with acute respiratory failure [19]. The aim of this study was to test whether our correction method could improve the accuracy in estimating ΔPpl compared to raw ΔCVP values and whether or not it could be used in pediatric patients who have spontaneous breaths during mechanical ventilation.

## Methods

### Study design and patient selection

This prospective study was performed in the pediatric intensive care unit of a tertiary children's hospital. The study was approved by the Institutional Review Board of Osaka Women's and Children's Hospital (February 2017, approval number 955). The requirement for written informed consent was waived by the institutional review board. Patients were considered for inclusion in the study if they were younger than 18 years, with sinus rhythm, were not supported with high-dose catecholamines (more than 0.05 mcg/kg/min of epinephrine equivalent), were mechanically ventilated under spontaneous breathing with a positive end-expiratory pressure of <10 cmH$_2$O, had a central venous catheter (CVC) inserted via the internal jugular vein, and had an esophageal balloon catheter placed for clinical purposes between

March 2017 and June 2017. Patients in whom correct positioning of the esophageal balloon catheter was not ensured were excluded.

## Setting for measurement and recording

The tip of the CVC was confirmed to be in the superior vena cava by chest radiography. The pressure transducer for CVP measurement was leveled at the mid-axillary line. Airway pressure (Paw) was measured at the junction of the respirator circuit and the endotracheal tube. An esophageal balloon catheter (AVEA™ ventilator Pes monitoring tube, IMI, Saitama, Japan) was inserted into the mid-lower third of the thoracic esophagus via the nasal route. Pes was measured in the supine position as follows: first, the balloon was completely deflated by applying negative pressure before each measurement of Pes; the balloon was then inflated with 0.5 mL of air and finally deflated to the target volume of 0.3 mL. Correct positioning of the esophageal balloon catheter was confirmed using the occlusion test, in which changes in Pes and Paw (ΔPes and ΔPaw, respectively) were measured while the patient was breathing spontaneously against a closed airway [7, 20]. The catheter position was deemed correct when the ratio of ΔPes to ΔPaw was between 0.8 and 1.2 during an occlusion test. We adjusted the body position, the length of the balloon insertion, and the amount of air in the balloon if the targeted ratio was not obtained. If such attempts were not successful within 30 minutes, the patient was then excluded. CVP, Pes, and Paw were displayed simultaneously on a bedside monitor (BSM-6701, Nihon Kohden, Tokyo, Japan) that used pressure transducers of the same model (pediatric TruWave pressure monitoring transducer, Edwards Lifesciences, CA, USA). Data were automatically transferred to and recorded in an electronic medical chart system (GAIA, Nihon Kohden) every 0.004 s using digital signals. The collected data were then exported to an Excel spreadsheet (Microsoft Excel, Microsoft Corporation, Redmond, WA, USA) for subsequent off-line analysis. Because the Paw, Pes, and CVP waveforms have cardiogenic oscillations, measurements taken at the bottom of the "y" descent or the bottom of the "x" descent when the "y" descent could not be identified (Fig 1). All measurements were performed with level -1 sedation on the State Behavioral Scale.

## Measurement and comparison of variables that reflect ΔPpl

First, we measured and calculated variables that reflect ΔPpl, that is, ΔPes, ΔCVP, and ΔPpl calculated using a corrected ΔCVP (cΔCVP-derived ΔPpl). To calculate the cΔCVP-derived ΔPpl, an occlusion test was performed to obtain the ratio of ΔPaw to ΔCVP (Fig 1A). This ratio was expressed as "k" and was presumably similar to the ratio of ΔPpl to ΔCVP because ΔPaw should be equal, or at least close, to ΔPpl during airway occlusion, unless there is severe chest wall distortion or air-trapping [18, 21]. After 5 min of stabilization under each ventilator setting, we measured ΔPes and ΔCVP of the same breath under 10, 5, and 0 cmH$_2$O of pressure support (PS) (Fig 1B). The other ventilator settings were unchanged during the measurements. Assuming the ratio of ΔPpl to ΔCVP during the occlusion test and during mechanical ventilation to be similar, cΔCVP-derived ΔPpl can be expressed as follows:

$$cΔCVP-derived\ ΔPpl = k × ΔCVP$$

Next, we examined the relationship between cΔCVP-derived ΔPpl and ΔPes. Given that ΔPes is widely accepted as a gold standard surrogate for ΔPpl, we used ΔPes as a reference value for ΔPpl. The correlation of ΔCVP and cΔCVP-derived ΔPpl with ΔPes at each PS level was also compared. The regression coefficients were also calculated.

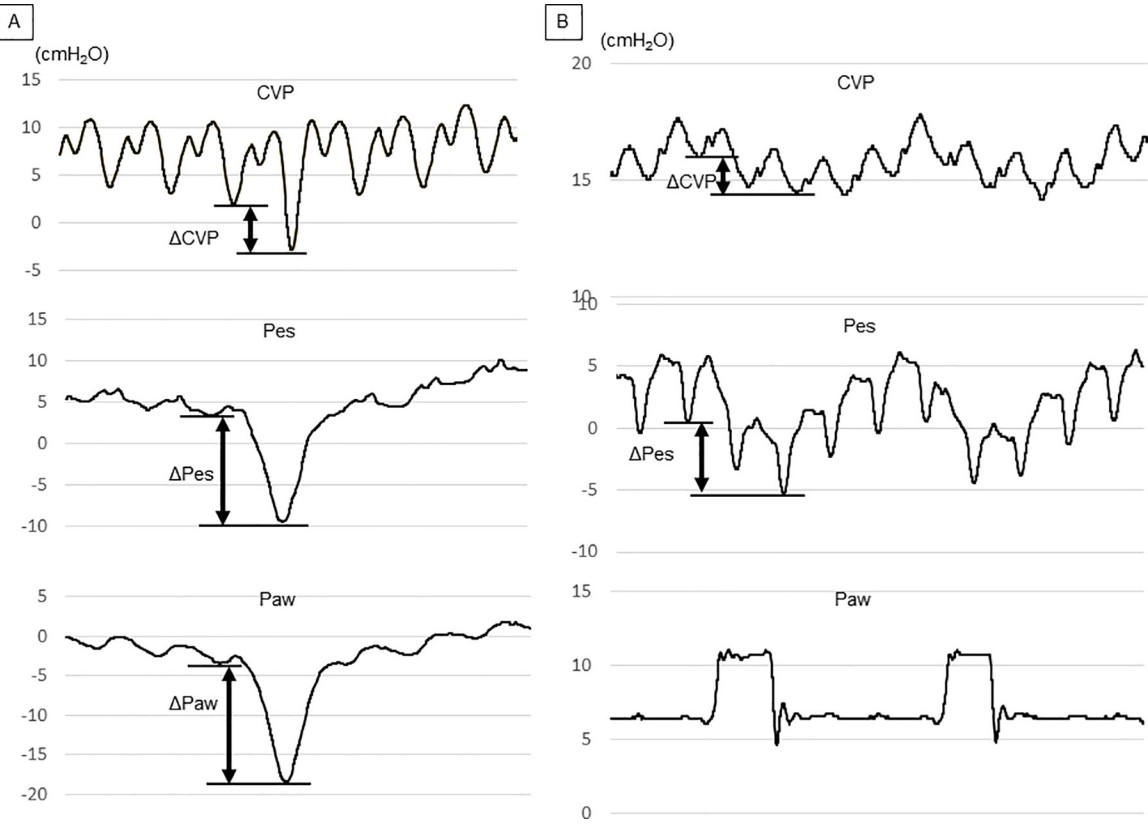

**Fig 1.** Pressure waveforms for CVP, Pes, and Paw during an occlusion test (A) and during assisted and unassisted spontaneous breathing under ventilatory support (B). (A) During an occlusion test, the three waveforms fluctuate in a similar manner. ΔPaw should be close to ΔPes during an occlusion test, provided that the position of the esophageal balloon catheter is correct. In Fig 1, for example, ΔPes was 12.9 $cmH_2O$ and ΔPaw was 15.2 $cmH_2O$, leading to a ΔPes to ΔPaw ratio of 0.85. The ratio of ΔPaw to ΔCVP obtained during the occlusion test was expressed as "k." (B) ΔCVP and ΔPes were measured during assisted spontaneous breathing under mechanical ventilation. The ΔPpl was then calculated by multiplying k of the same patient by ΔCVP. CVP: central venous pressure; ΔCVP: change in central venous pressure; Pes: esophageal pressure; ΔPes: change in esophageal pressure; Paw: airway pressure; ΔPaw: change in airway pressure; ΔPpl: change in pleural pressure.

### Statistical analysis

Continuous variables are presented as mean ± standard deviation. We sought to determine whether there was a linear relationship between cΔCVP-derived ΔPpl and ΔPes. Repeated measurement correlation was performed because multiple measurements were taken for individual patients [22]. To assess the accuracy and precision of predicting ΔPes using ΔCVP and cΔCVP-derived ΔPpl, we performed descriptive statistics on the difference between the ΔPes and the two methods (ΔCVP and ΔCVP-derived ΔPpl). The correlation of ΔCVP and cΔCVP-derived ΔPpl with ΔPes at each PS level was tested using Pearson's product-moment correlation coefficient. Repeated measurement correlations were performed using R (The R Foundation for Statistical Computing, Vienna, Austria). Other statistical analyses were performed using EZR version 1.36 (Saitama Medical Center, Jichi Medical University, Saitama, Japan), which is a graphical user interface for R. A p-value $< 0.05$ was considered statistically significant.

### Results

Fourteen patients were enrolled in the study. After the exclusion of 6 patients in whom correct positioning of the esophageal balloon could not be confirmed during the occlusion test, eight

**Table 1. Patient characteristics.**

|  | case 1 | case 2 | case 3 | case 4 | case 5 | case 6 | case 7 | case 8 |
|---|---|---|---|---|---|---|---|---|
| age (month) | 9 | 3 | 0 | 2 | 3 | 4 | 7 | 10 |
| weight (kg) | 6.5 | 3.3 | 3.8 | 3.6 | 4.1 | 3.3 | 6.2 | 6.3 |
| sex | female | male | male | male | female | male | male | male |
| diagnosis | cerebral infarction | CAVC | TGA | CAVC | DORV | CAVC | Cardio-myopathy | MR |
| reason for intubation | pneumonia/pulmonary edema | operation | operation | operation | operation | operation | shock | cardiac failure |
| length of mechanical ventilation (days) | 8 | 6 | 5 | 5 | 17 | 3 | 23 | 6 |
| length of ICU stay (days) | 12 | 19 | 11 | 13 | 20 | 10 | 34 | 10 |
| days from intubation to study enrollment (days) | 7 | 4 | 4 | 4 | 16 | 3 | 16 | 3 |

CAVC, common atrioventricular canal; TGA, transposition of the great arteries; DORV, double-outlet of right ventricle; MR, mitral regurgitation.

patients were included in the analysis. Six of these eight patients were male. The average age of the patients was 4.8 months. The patient characteristics are shown in Table 1, while the circulatory and respiratory parameters during study enrollment are shown in Table 2.

The respective mean and standard deviation values for the three variables that reflected ΔPpl (ΔPes, ΔCVP, and cΔCVP-derived ΔPpl) were − -6.7 ± 4.8, − -2.6 ± 1.4, and − -7.3 ± 4.5 cmH$_2$O, respectively. The difference of cΔCVP-derived ΔPpl to ΔPes tended to be smaller than that of ΔCVP to ΔPes in all settings (-0.1 ± 1.5 vs. 3.1± 3.5 cmH$_2$O in PS10, -0.7 ± 3.3 vs. 4.5± 3.9 cmH$_2$O in PS5, and -1.0 ± 3.4 vs 4.7± 4.4 cmH$_2$O in PS0).

The repeated measures correlation between cΔCVP-derived ΔPpl and ΔPes showed that cΔCVP-derived ΔPpl had good correlation with ΔPes (r = 0.84, p< 0.0001) (Fig 2). The correlation of cΔCVP-derived ΔPpl with ΔPes was not perfect but was slightly stronger than the correlation of ΔCVP with ΔPes at all PS levels (Fig 3). In addition, the regression coefficients of cΔCVP-derived ΔPpl and ΔPes were closer to 1 than those of ΔCVP and ΔPes for all PS levels (1.11 vs. 2.38 in PS10, 0.72 vs 2.56 in PS5, and 0.89 vs 2.56 in PS0).

**Table 2. Patient parameter.**

|  |  | case 1 | case 2 | case 3 | case 4 | case 5 | case 6 | case 7 | case 8 |
|---|---|---|---|---|---|---|---|---|---|
| heart rate (/min) |  | 100 | 140 | 135 | 128 | 130 | 138 | 126 | 116 |
| mean arterial pressure (mmHg) |  | 90 | 52 | 56 | 58 | 58 | 56 | 54 | 63 |
| Blood gas analysis at study enrollment | pH | 7.44 | 7.48 | 7.47 | 7.43 | 7.39 | 7.42 | 7.44 | 7.46 |
|  | PaO$_2$ (mmHg) | 106 | 88 | 159 | 152 | 108 | 135 | 132 | 99 |
|  | PaCO$_2$ (mmHg) | 37 | 40 | 38 | 40 | 47 | 46 | 42 | 42 |
|  | lactate (mg/dL) | 7.0 | 9.0 | 8.0 | 6.0 | 7.0 | 8.0 | 5.0 | 7.0 |
| P/F ratio (mmHg) |  | 505 | 176 | 265 | 304 | 216 | 300 | 528 | 330 |
| Respiratory rate (/min) |  | 22 | 28 | 25 | 30 | 25 | 32 | 28 | 20 |
| Tidal volume (mL) |  | 54 | 31 | 26 | 27 | 28 | 47 | 62 | 64 |
| minute volume (L) |  | 1.2 | 0.87 | 0.65 | 0.81 | 0.7 | 1.50 | 1.74 | 1.28 |
| F$_I$O$_2$ |  | 0.21 | 0.5 | 0.6 | 0.5 | 0.5 | 0.45 | 0.25 | 0.3 |
| PEEP (cmH$_2$O) |  | 5 | 6 | 7 | 6 | 6 | 6 | 5 | 6 |
| CVP (cmH$_2$O) |  | 4 | 11 | 7 | 7 | 9 | 6 | 3 | 11 |
| ΔCVP during occlusion test(cmH$_2$O) |  | 8.6 | 5.9 | 6.3 | 8.2 | 2.4 | 2.4 | 10.4 | 3.1 |
| k |  | 1.59 | 2.25 | 2.59 | 1.83 | 3.79 | 2.31 | 3.66 | 3.58 |

PEEP, positive end-expiratory pressure; CVP, central venous pressure; k, ratio of ΔPaw to ΔCVP obtained during an occlusion test.

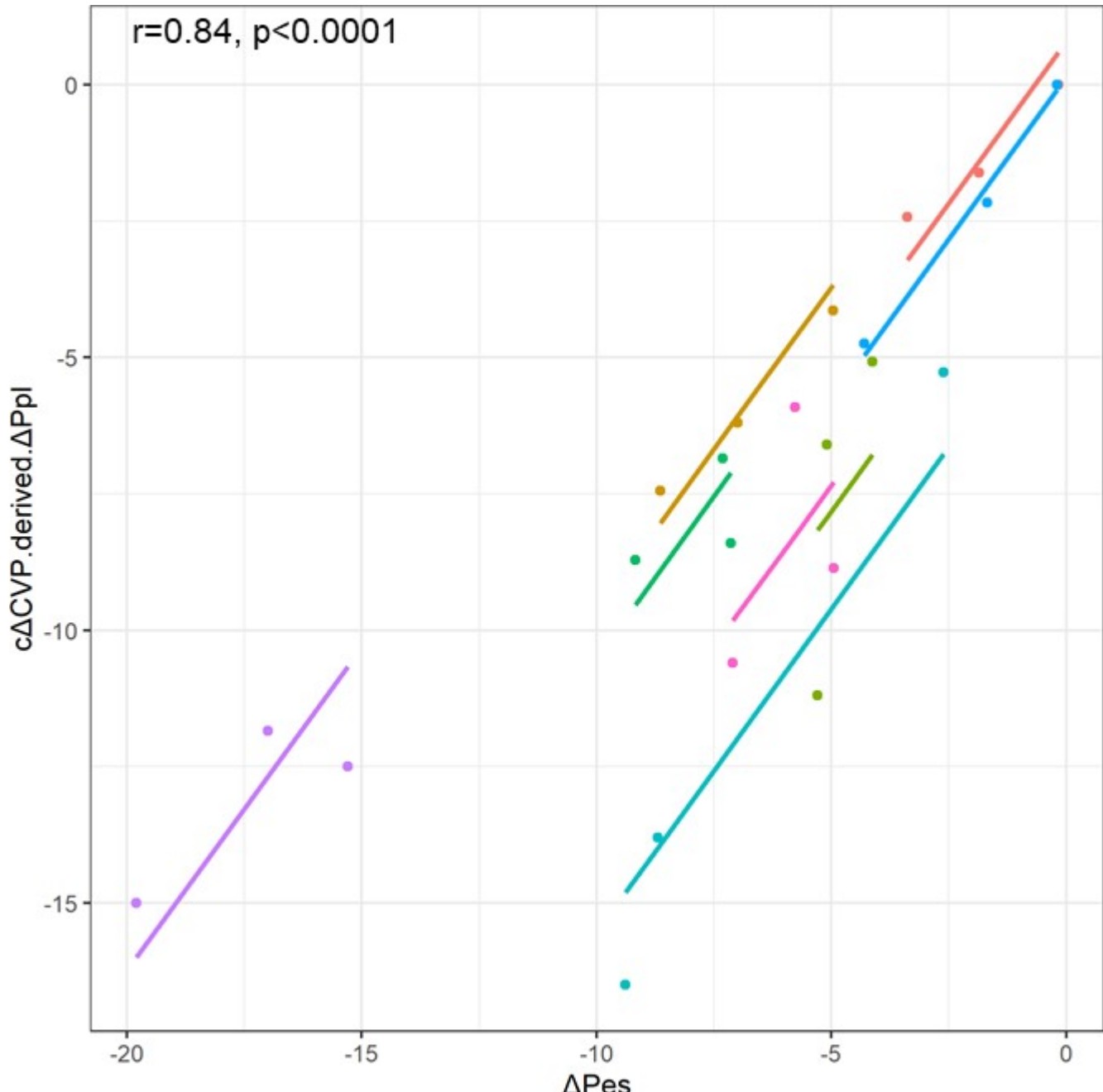

**Fig 2. Scatter plots for the repeated measures correlations between cΔCVP-derived ΔPpl and ΔPes.** For comparison, individual data are colored differently. The dots represent the data for each patient, and the corresponding lines represent the linear relationships for each patient. ΔCVP, change in central venous pressure; ΔPpl, change in pleural pressure; ΔPes, change in esophageal pressure; cΔCVP-derived ΔPpl, ΔPpl calculated using a corrected ΔCVP.

## Discussion

We have previously reported that ΔPpl can be estimated without an esophageal balloon catheter using ΔCVP that is adjusted with a simple correction method in mechanically ventilated, paralyzed pediatric patients with acute respiratory failure [19]. However, it was not known whether our method could be used in pediatric patients with assisted and unassisted

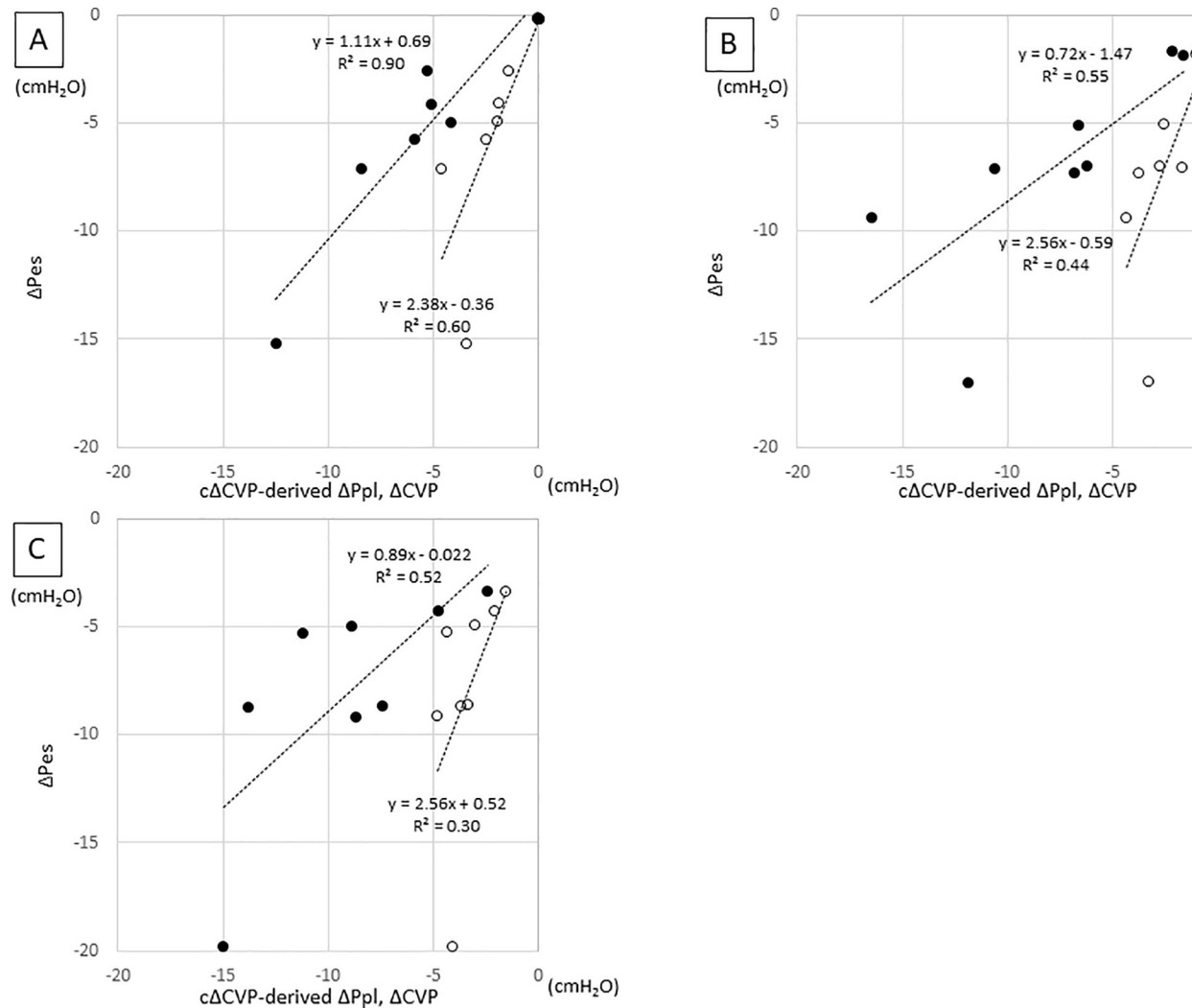

**Fig 3. Relationship between cΔCVP-derived ΔPpl or ΔCVP and ΔPes.** A: PS10, B: PS5, C: PS0. In each figure, filled circle represent cΔCVP-derived ΔPpl and open circle represent ΔCVP.

spontaneous breathing, given that respiratory efforts translate into ΔPpl. In this preliminary study, we have shown that using our method, ΔPpl can be estimated reasonably accurately by ΔCVP in assisted and unassisted spontaneously breathing children. Our preliminary data also indicate that the correction method seemed to be able to estimate ΔPes more accurately than the method using raw ΔCVP.

Both excessive and insufficient muscle loading by a mechanical ventilator have been shown to be associated with VIDD and P-SILI [1–6]. VIDD is associated with poor outcomes, such as prolonged mechanical ventilation and extended stays in the intensive care unit [1, 2]. On the other hand, P-SILI may be the hidden cause of lung damage, even with low tidal volume and low plateau pressure [3, 4]. Therefore, it is important to monitor respiratory effort and titrate ventilator settings to keep it at an appropriate level. Specifically, pressure generated by the respiratory muscles between 5 and 10 $cmH_2O$ was recommended as a desirable respiratory effort during partial ventilatory support [6, 23]. In general, respiratory effort is estimated by measuring Pes, which is a surrogate of Ppl [7].

However, there measurement of Pes is complicated by technical issues, such as those related to positioning the catheter correctly and interpreting the obtained values [8, 9]. For this reason, esophageal balloon catheters were inserted in only 0.8% of patients with acute respiratory distress syndrome in a recent study [24]. However, it is not uncommon for a mechanically ventilated patient with respiratory failure to have a CVC inserted via the internal jugular vein. Tidal swings in CVP have been shown to reflect ΔPpl during the respiratory cycle [11, 13]. In a recent editorial regarding respiratory treatment of COVID-19, in the absence of an esophageal catheter, the use of the swings of CVP as a surrogate measure for the work of breathing was recommended [25]. Although ΔCVP was correlated with ΔPes in previous studies [11, 13], many studies have shown that ΔCVP did not usually reflect the exact value of ΔPes [10, 12, 14–18]. Lung volume, chest wall elastance, chest wall distortion, and volume status including CVP and air trapping may affect the relationships between ΔPpl, ΔPes, and ΔCVP [12, 21, 26–29]. As a result, the reported values of ΔPes/ΔCVP were not consistent and, more importantly, varied widely among individuals [10–17]. To overcome this problem, we used the ratio of ΔPpl to ΔCVP during an occlusion test to correct the raw ΔCVP values and estimate ΔPpl more accurately than when simply using ΔCVP.

Several requirements are necessary when attempting to apply our method. It requires that the CVC be placed in the superior vena cava. Cardiac pathophysiology, including arrhythmia and tricuspid regurgitation, may render the use of ΔCVP invalid as a method for estimation of ΔPpl. However, our method of using ΔCVP to estimate ΔPpl has several advantages over the esophageal balloon catheter method. A CVC may be inserted in pediatric patients with respiratory failure for several reasons, including difficult vascular access and administration of vasoactive medications, whereas esophageal balloon catheters are not widely used, even in tertiary children hospitals. Moreover, even if esophageal balloon catheters are available, some patients may fail the occlusion test. Similarly, even in studies of adults in institutions accustomed to using Pes, it was reported that 37% of all recordings did not pass the occlusion test and were ultimately excluded [30]. In such cases, our method of using the ΔCVP may be more reliable for estimating ΔPpl than the esophageal balloon catheter method.

Furthermore, our method is minimally invasive compared to the insertion of an esophageal balloon, provided that a CVC has been inserted for other clinical purposes. Even though our method of estimating ΔPpl is not perfect, our method seems to be more accurate than when using raw ΔCVP data (Fig 3). Therefore, our method could still be used as a screening tool to select patients who would benefit from monitoring of Pes.

In both our previous and present studies, more than 40% of the cases (5/12 and 6/14) did not pass the occlusion test despite the seemingly correct radiographic position of the esophageal balloon catheter [19]. There were several possible reasons for this. First, in infants, because the chest wall is more compliant than in adults and inspiratory efforts easily distort the chest wall inward direction, it was shown that ΔPes is not necessarily equivalent to ΔPaw (mean Ppl swings) during occlusion test in the presence of distortion [21]. Second, we used a balloon catheter instead of a liquid-filled nasogastric catheter, which may be more accurate in small infants [27, 31]. Third, the size and volume of the balloon may not have been appropriate for infants [32]. However, this balloon catheter is currently the only commercially available equipment for measuring the Pes of infants in Japan. Fourth, since there were many post-cardiac surgery patients in our patient group, it is possible that hematomas, adhesions in the thoracic cavity, and indwelled pleural catheters (with negative pressure of 5–7 cmH$_2$O) may have influenced the relationship between ΔPao and ΔPes during occlusion. Finally, the large distortion of the thorax in neonates by inspiratory efforts may affect the relationship between ΔPao and ΔPes during occlusion [21].

Our study has several limitations. First, the number of included patients was limited. Therefore, we could not statistically prove that our method was accurate. Second, all patients included in this study were infants, although we had intended to include pediatric patients aged up to 18 years. Therefore, our method needs to be validated in a larger and more diverse population, such as adult ARDS patients. Thirdly, we selected the bottom of "x" or "y" descent of the cardiogenic oscillations to measure ΔCVP and ΔPes in this study. Selecting other points may improve the accuracy of our estimation method [15]. Fourth, we assumed that k (ΔPpl to ΔCVP ratio) obtained during the occlusion test was similar to that obtained during mechanical ventilation. However, to be precise, this assumption may not always be true. Because PEEP affects lung volume and lung volume affects ΔPpl/ΔCVP [28], ΔPpl/ΔCVP during the occlusion test with no PEEP is different from that during mechanical ventilation with PEEP. Moreover, the pattern of blood flow into the right atrium may not be the same during airway occlusion and mechanical ventilation [15], which may affect the pressure and compliance of the right atrium and, as a result, ΔPpl/ΔCVP may also be affected. However, the ratio of ΔPes/ΔCVP to k during occlusion at PS10, PS5, and PS0 were acceptable (0.86±0.31, 0.98±0.27, and 0.95±0.34, respectively) in this study.

## Conclusions

In conclusion, our method of estimating ΔPpl without an esophageal balloon catheter using ΔCVP during the respiratory cycle and correcting the raw ΔCVP value may be reliable when used among assisted and unassisted spontaneous breathing pediatric patients. Further validation studies are warranted in a larger and more diverse patient population.

## Supporting information

**S1 Data.**
(XLSX)

## Author Contributions

**Data curation:** Nao Okuda, Miyako Kyogoku.

**Formal analysis:** Nao Okuda, Yu Inata, Muneyuki Takeuchi.

**Investigation:** Nao Okuda, Kanako Isaka, Kazue Moon, Takeshi Hatachi, Yoshiyuki Shimizu.

**Supervision:** Muneyuki Takeuchi.

**Writing – original draft:** Nao Okuda.

**Writing – review & editing:** Yu Inata, Muneyuki Takeuchi.

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
