## [Decision Letter · Decision Letter 0]

11 Sep 2020

PONE-D-20-20085

Estimation of change in pleural pressure in spontaneously breathing pediatric patients using fluctuation of central venous pressure: A proof-of-concept study

PLOS ONE

Dear Dr. Takeuchi,

Thank you for submitting your manuscript to PLOS ONE. After careful consideration, we feel that it has merit but does not fully meet PLOS ONE’s publication criteria as it currently stands. Therefore, we invite you to submit a revised version of the manuscript that addresses the points raised during the review process.

- Please clarify the reasons for excluding such a high number of patients (in proportion).

- Debate about the optimal CVP trace point to be used for dCVP measurement. Otherwise data should be recalculated.

- Please report the K value also for dPes and discuss which factors influence the K "transmission" factor (chest wall? baseline PCV? PEEP?)

- Please discuss the impact of chest wall compliance on your results, and if it is related with thoracic/cardiac surgery or different PS levels.

- As noted by the reviewers, it is not possible to pool more measurements from the same patients together. Analysis and graphs need to be changed accordingly. Also consider RM Anova analysis to explore differences.

We look forward to receiving your revised manuscript.

Kind regards,

Andrea Coppadoro

Academic Editor

PLOS ONE

Journal Requirements:

2. Please state in your methods section when you conducted this study.

Reviewers' comments:

Reviewer's Responses to Questions

**Comments to the Author**

1. Is the manuscript technically sound, and do the data support the conclusions?

Reviewer #1: Yes

Reviewer #2: Yes

Reviewer #3: Partly

Reviewer #4: Yes

2. Has the statistical analysis been performed appropriately and rigorously? 

Reviewer #1: Yes

Reviewer #2: Yes

Reviewer #3: Yes

Reviewer #4: Yes

3. Have the authors made all data underlying the findings in their manuscript fully available?

Reviewer #1: Yes

Reviewer #2: Yes

Reviewer #3: Yes

Reviewer #4: Yes

4. Is the manuscript presented in an intelligible fashion and written in standard English?

Reviewer #1: Yes

Reviewer #2: Yes

Reviewer #3: Yes

Reviewer #4: Yes

5. Review Comments to the Author

Reviewer #1: please,the authors should omitt to say" proof of the concept" in the title,as this concept is known since several years and dacades

the authors should add to table one the values of delta CVP .this would allow to compute the Paw during the occlusion

Pleas, discuss the possible reasons of the great k variabilty

Reviewer #2: RE: PONE-D-20-20085

This is an interesting paper that describes an improved method of estimating pleural pressure swings (delta Pes) in spontaneously breathing infants by simultaneously measuring swings in CVP (delta CVP). Purpose is to assess use of the CVP as a substitute for recording inspiratory efforts during weaning off ventilation. The authors are to be commended in attempting to compensate for potential technical difficulties by performing the occlusion test and simultaneously measuring swings in airway pressure (delta Paw) with CVP swings. The ratio of the change in Paw and CVP was expressed as a correction factor, k. The “corrected” swing in CVP was then compared to simultaneously recorded esophageal pressure measurements with an air-filled esophageal balloon-catheter system. The authors found that correlation between Pes and CVP swings was fair with acceptable bias.

The authors argue that using the CVC to measure CVP changes is more convenient, comfortable and perhaps less likely to encounter technical difficulties than with esophageal balloon systems.

General comments:

The authors have given serious consideration to the technical issues and cite appropriate literature supporting their case, although they could have searched deeper into the literature (see below). There are also other technical issues they need to address. The paper is well-written, with appropriately constructed background, hypothesis, methods, results and discussion sections.

Specific comments:

1. The authors should explain their choice of using an air-filled balloon catheter system instead of a saline or water-filled catheter system, which has been the more common method used in infants. They should refer to the work of Asher et al [Measurement of pleural pressure in infants. J Appl Physiol 1982; 52(2): 491-494]. Air-filled balloon catheters have a reduced frequency response, resulting in reduced delta Pes, with underestimation of inspiratory efforts, particularly in rapidly breathing infants. This finding was also shown by Beardsmore et al [J Appl Physiol 1980; 49(4):735-742] who used air-filled balloons to measure lung mechanics in their patients. A poor frequency response is also associated with a phase lag, which, if small, would lead to a measured swing in delta Pes that would be greater than delta Ppl. Water-filled catheters are also said to be more comfortable for babies. Finally, catheter thickness and balloon size also make a difference (Beardsmore et al) – but I assume the authors were limited to the catheter system provided by AVEA.

2. P. 6: The authors should cite the original literature describing the modern technique for recording esophageal pressure (Milic-Emili et al. Improved technique for estimating pleural pressure from esophageal balloons. J Appl Physiol 1964; 19(2): 207-211) and the occlusion test itself (Baydur et al. A simple method for assessing the validity of the esophageal balloon technique. Am Rev Respir Dis 1982; 126:788-791). Akoumianaki et al (ref. 7) merely applied an already established technique to ventilated patients; they were not the original investigators. And their study had the same technical limitations as this study.

3. P. 7, first paragraph: It is important to make certain that Paw, Ppl, and CVP are measured at the same end-expiratory volume (which is hard to do in ventilated patients unless flow and volume are accurately recorded). Any increase or decrease in lung volume will over- or underestimate pressure swings. The authors do not describe monitoring flow and volume during their experiments.

4. P. 7: The authors made measurements at different pressure support levels (line 6) – this will alter end-expiratory volume, resulting in changes in pressure swings as described above. In addition, as the authors point out later, less pressure support should result in greater chest wall distortion because of greater diaphragmatic contraction, leading to

greater discrepancy between Paw and Ppl and CVP changes. Contrary to what the authors assumed, such changes do not produce “a steady state” (line 8).

5. P. 7: Again, same problem: Authors state that changes in Pes are assumed to be the same as changes in Ppl – not true if there is chest wall distortion (see below) or increased airway resistance. Changes in Pes and secondarily cCVP can only be used as a gross estimate of pleural pressure changes.

6. P. 8: Results: Please explain why 6 of 14 patients were excluded. Was it just because of chest wall distortion? Perhaps the occlusion test ratio would have been closer to unity had water-filled catheters been used. Authors should comment on the exclusions.

7. P. 9: Table 1: Six of 8 patients underwent cardiac surgery which can cause distortion of pleural pressure distribution because of pleural effusions and adhesions. It is likely that these patients had pleural catheters in place to drain blood and fluid. This may have been the reason why changes in Ppl did not reflect changes in Pes and CVP. Authors should describe if pleural catheters were in place and if they were attached to suction – this would really affect their measurements.

8. P. 9: Another point --- if indeed patients had pleural catheters in place, why not directly compare the changes in Ppl with changes in CVP? This would have been a lot easier than making corrections for the CVP.

9. Figs. 2 and 3: I counted 22 data points in both plots. How many recordings did the authors obtain in each patient? Are the data points evenly distributed amongst the 8 patients? How did the authors know which data points to include? Was there selection bias and some data points excluded?

10. Again, figures: The authors should explain the outliers in the correlation and Bland-Altman plots. There are at least 4, possibly more outliers. The authors should explain why these data points varied so much from the main group. Possible explanations could include underlying clinical conditions (pneumonia, heart failure, pleural effusions), the respiratory rate, pressure support level and intravascular volume affecting the CVP.

Reviewer #3: The authors presented a manuscript on CVP swing as a simple and non-invasive assessment of pleural pressure in spontaneously breathing children.

The topic is very interesting and the clinical impact of such finding could be immediate and appreciated. The first field of application could be the assessment of respiratory efforts during spontaneous ventilation, where PES is scarcely measured and strong CVP swing could lead to an early identification of patients at high risk of self induced lung injury.

Patients admitted to units where esophageal probe is not available and those with a contra-indication to the positioning of an esophageal balloon (such as those with facial trauma or after esophageal surgery) could benefit from the CVP analysis proposed by authors.

Even if the manuscript is interesting, I have major concerns regarding the study. Here are my comments:

MAJOR COMMENTS:

Can you better define "haemodynamic stable" patient? How were patients with arrhythmia considered? Were they excluded?

In your previous manuscript (https://doi.org/10.1007/s10877-019-00368-y) you stated that "Pressure values at the peak of the cardiogenic oscillations were used for calculation" while in this article you stated that pressure values at the bottom of the cardiogenic oscillations were used in the calculations". Why did you change your method?

During an occlusion test Pes and Paw waveforms should fluctuate in a similar manner and ΔPaw should be equal to Δpes (accepted discrepancy ±20%ccordingly to Baydur occlusion test). In Figure 1A you reported the pressures waveforms recorded during an occlusion test: Δpaw is significantly greater than Δpes (it seems ratio of Δpes to ΔPaw 0,7). Can you explain this?

The aim of your study is to estimate Ppl swings using fluctuations of CVP.

ΔPes is a surrogate of ΔPpl.

You calculated k as the ratio of Δpaw to ΔCVP obtained during an occlusion test. Why did you use Δpaw instead of Δpes?

ΔPaw isn’t exactly equal to Δpes and therefore k would be different if Δpes was the numerator instead of ΔPaw. Can you perform calculation with k = Δpes/ ΔCVP?

Can you add to figure 1B the Paw wave and highlight the chaqnge in Paw during breathing?

k value (range 1.59-3.79) varies a lot between subjects, how do you explain this heterogeneity?

Mean k value is between 2-3 that means ΔPaw is 2-3 times greater than Δpes during an occlusion test . How do you explain this ratio? It could be that the pressure drop in the chest increases the venous return, with the heart not handling it and the PVC rising (so that the PVC delta is relatively low). Interestingly, the only one non-cardiopatic patient has the lowest k value: pleas comment.

You excluded more than 42% of scrreened children because esophageal balloon position could not be confirmed during the occlusion test. It’s a very high rate, what’s your explanation? Didn’t position and inflation adjustment resolve the discordance?

In children successfully enrolled you measured Δpes and ΔCVP under 10, 5 and 0 cmH2O of pressure support obtaining presumably a total of 24 measurements.

1. Babies who receive pressure support are in assisted spontaneous breathing patients. You obtained 16 measurements during assisted ventilation and only 8 during unassisted spontaneous breathing. Please make it explicit in the title and text

2. You analyzed all obtained measurements together: it can be methodologically incorrect. You have to analyze measurements obtained in different settings separately because pressure support magnitude can differently influence Δpes and ΔCVP. Please provide three different analysis: pressure support 10, 5 an 0.

3. During data collection did you change only pressure support or did you modify other ventilator parameters (PEEP, trigger,…).

4. Can you add a table to illustrate how other parameters (respiratory rate, paO2, paO2/FiO2, paCO2, pH, heart rate, blood pressure, capillary refill, sedation level) modified during pressure support modification?

For each patient add ventilator parameters (Pplateau, mean airway pressure, PEEP, tidal volume, respiratory rate, minute volume), Δpes, Δpaw, ΔCVP, cΔCVP-derived Δppl, ΔPes-derived plateau PL cΔCVP-derived plateau PL recorded at the moment of data collection.

Sample size: the small number of patients enrolled reduces the attendibility of statistical analysis. Bland-Altmann is not validated for analysis of so few repeated measures. Discuss this item. Furthermore agreement assessed with the Bland-Altman analysis should be described as the median difference (bias) and 2.5th and 97.5th percentiles (95%-limits of agreement [LoA] (Bland JM, Altman DG. Statistical methods for assessing agreement between two methods of clinical measurement. Lancet 1986;1:307-310)

Several studies (Colombo J et al. Detection of strong inspiratory efforts from the analysis of central venous pressure swings: a preliminary clinical study, Minerva Anestesiologica, in press 10.23736/S0375-9393.20.14323-2; Hedstrand U, Jakobson S, Nylund U, Sterner H. The concordance of respiratory fluctuations in oesophageal and central venous pressures. Ups J Med Sci 1976;81:49-53; Flemale A, Gillard C, Dierckx JP. Comparison of central venous, oesophageal and mouth occlusion pressure with water-filled catheters for estimating pleural pressure changes in healthy adults. Eur Respir J 1988;1:51-57; Chieveley-Williams S, Dinner L, Puddicombe A, Field D, Lovell AT, Goldstone JC. Central venous and bladder pressure reflect transdiaphragmatic pressure during pressure support ventilation. Chest 2002;121:533-538; Bellemare P, Goldberg P, Magder SA. Variations in pulmonary artery occlusion pressure to estimate changes in pleural pressure. Intensive Care Med 2007;33:2004-2008; Verscheure S, Massion PB, Gottfried S, Goldberg P, Samy L, Damas P, et al. Measurement of pleural pressure swings with a fluid-filled esophageal catheter vs. pulmonary artery occlusion pressure. J Crit Care 2017;37:65-71) report a poor agreement but a positive and significant correlation between ΔCVP and ΔPES: even if ΔCVP and ΔPES were not always the same, smaller or larger ΔCVP generally reflected smaller or larger ΔPES. Starting from your data and the manuscripts above you shold better discuss this theme and explain what your work add to previous data

MINOR:

In small children chest x-ray cannot confirm the exact position of CVC tips immediately above the SVC-right atrium junction. This could produce a bias in CVP measurements. I think electrocardiogram (ECG)-guided technique could be a more precise technique. Why didn't you used this approach?

Are all patients in respiratory weaning or some of them are in an acute phase of respiratory illness? Pleas add this information in population description

Most patients are post-cardiac surgery ones: was diaphragmatic function normal in all of them?

Please add the correlation test you used to performed analysis

In table 1 add patients’ age, reason to PICU admission, initial severity of disease, days from icu admission to study enrollment

Add table to describe main characteristics of the study population at study entry: FiO2, Arterial pH, Arterial CO2 tension, Arterial O2 tension, Arterial O2 saturation, PaO2:FiO2, Heart rate, Mean arterial pressure, Central venous pressure, Central venous O2 saturation, Lactate, Urinary output, Vasopressors, Clinical evaluation of volemic status (Hypovolemic, Normovolemic, Hypervolemic)

Revise English and grammar

CONCLUSIONS

The work is of some interest but it has some methodological limitations (also related to the statistical analysis) that needs to be addressed. I think, due to the small sample size, the most appropriate article type for the manuscript is a preliminary report.

Reviewer #4: In this paper, the authors describe a methodology by which to measure pleural pressure swings during spontaneous ventilation in pediatric patients by using the respiratory swings in central venous pressure(CVP) measurements.

We and others have been looking for such a method to answer important clinical questions which touch upon the respiratory and circulatory status of many intensive care patients and therefore I agree the authors are posing an important clinical query.

Their initial cohort comprised 14 patients who had been instrumented, for others reasons other than the performance of these measurements, with a central venous catheter in the superior vena cava and an esophageal catheter-balloon apparatus. Their results include those only from only 8 patients, with a mean age of 4.8 months, as the position of the esophageal balloon could not be confirmed by airway occlusion in 6.

In calculating the fall in CVP, they adjusted that value to a constant (k) determined by the ratio between the raw fall in CVP to the simultaneous fall in airway pressure during an airway occlusion assuming that ratio should be the same as that between the fall in CVP to that of Peso - although those latter values were not reported.

They found a correlation of 0.56 and a bias on Bland-Altman analysis of 0.53. They concluded that the fall in CVP during spontaneous respiration can be used as a reasonably accurate assessment of pleural pressure changes.

Critique:

The use of this correction factor with which the authors derive CVP-derived Pl is imaginative and perhaps is responsible for the bias they found. It would be very helpful to report the ratios they also found in the fall in CVP to the fall in pleural pressures as measured by the fall in Peso. They mention that they should be the same as the ratio between CVP and airway but were they?

Furthermore, the authors recognize the limitations of the study and its generalizability given the few subjects and are cognizant of the possible limitations of using the fall in CVP at higher levels. Whereas their "indexation" should work at more modest CVP values, during an the airway occlusion manueuvre the diaphragms should not move caudal and propel blood from a charged splanchnic circulation into the right atrium which, during spontaneous ventilation, could further dilate the right atrium, increase its pressure, and make transmission of pleural pressures to the non-compliant right atrium even more problematic.

Finally, I am curious on how the authors measured both the CVP - to which landmark was the vascular transducer leveled - and how the fall in CVP was measured. In terms of the latter, it appears from the graph 1A, that the authors use the bottom of the "Y descent" as the land mark and I'm not sure that is appropriate; using the base of the "a" wave or, failing that, the mean CVP would appear superior.

6. PLOS authors have the option to publish the peer review history of their article (what does this mean?). If published, this will include your full peer review and any attached files.

Reviewer #1: No

Reviewer #2: No

Reviewer #3: **Yes: **Jacopo Colombo

Reviewer #4: No

---

## [Author Response · Author response to Decision Letter 0]

4 Jan 2021

PONE-D-20-20085

Estimation of change in pleural pressure in spontaneously breathing pediatric patients using fluctuation of central venous pressure: A proof-of-concept study

Dear Dr Andrea Coppadoro,

We are pleased to submit the revised version of our manuscript (PONE-D-20-20085), now entitled “Estimation of change in pleural pressure in assisted and unassisted spontaneous breathing pediatric patients using fluctuation of central venous pressure: A preliminary study.”

We have revised the manuscript according to the editors and reviewers’ comments. The parts of the text where the content has been corrected are highlighted in yellow. We hope that our paper will be suitable for publication in your journal. We look forward to hearing from you concerning your editorial decision. Yours sincerely,

Muneyuki Takeuchi

Email: mutake1017@gmail.com(MT) 

EQ-1. Please clarify the reasons for excluding such a high number of patients (in proportion). 

Thank you for pointing that out. As you noted, a high number of patients were excluded. 

The reason for excluding this high number of patients is that many of our patients did not pass the occlusion test, despite 30 minutes of effort to adjust the body position, the length of the balloon insertion, and the amount of air in the balloon. Even though one paper from another group showed a similar percentage of exclusion (Reference 29), possible reasons for the disagreement between ΔPaw and ΔPes during occlusion in our study include the following:

First, we used a balloon catheter instead of a liquid-filled nasogastric catheter, which may be more accurate than a balloon catheter (Reference 30). Our initial target group was children under 18 years of age; and we believe that the use of the balloon was common in that age group when we planned our study. Second, the size and volume of the Pes balloon that we used may not have been appropriate for infants (Reference 31). However, as we described in the methods, we used catheters that were inserted for clinical purposes. This balloon catheter is the only commercially available and approved equipment for measuring esophageal pressure in Japan. Third, since there were many post-cardiac surgery patients in our patient group, it is possible that hematomas and adhesions in the thoracic cavity may have influenced the relationship between ΔPao and ΔPes during occlusion. Fourth, it is possible that the large distortion of the thorax in neonates by inspiratory efforts may affect the relationship between Pes and Ppl (Reference 21). 

Therefore, we added the following sentences to the discussion:

P16 L265-

Similarly, even in studies of adults in institutions accustomed to using Pes, it was reported that 37% of all recordings did not pass the occlusion test and were ultimately excluded [29].

P17 L276

In both our previous and present studies, more than 40% of the cases (5/12 and 6/14) did not pass the occlusion test despite the seemingly correct radiographic position of the esophageal balloon catheter [19]. There were several possible reasons for this. First, in infants, because the chest wall is more compliant than in adults and inspiratory efforts easily distort the chest wall inward direction, it was shown that ΔPes is not necessarily equivalent to ΔPaw (mean Ppl swings) during occlusion test in the presence of distortion [21]. Second, we used a balloon catheter instead of a liquid-filled nasogastric catheter, which may be more accurate in small infants [27,30]. Third, the size and volume of the balloon may not have been appropriate for infants [31]. However, this balloon catheter is currently the only commercially available equipment for measuring the Pes of infants in Japan. Fourth, since there were many post-cardiac surgery patients in our patient group, it is possible that hematomas, adhesions in the thoracic cavity, and indwelled pleural catheters (with negative pressure of 5–7 cmH2O) may have influenced the relationship between ΔPao and ΔPes during occlusion. Finally, the large distortion of the thorax in neonates by inspiratory efforts may affect the relationship between ΔPao and ΔPes during occlusion [21]. 

And we also added the following sentence to the method:

P6 L104

We adjusted the body position, the length of the balloon insertion, and the amount of air in the balloon if the targeted ratio was not obtained. If such attempts were not successful within 30 minutes, the patient was then excluded.

EQ-2. Debate about the optimal CVP trace point to be used for dCVP measurement. Otherwise data should be recalculated.

We tried to always measure at the same point in a cardiac contraction, although it is not clear where in the CVP waveform is most correct to measure at. One previous study used the base of the “c” or “a” wave to measure the swing of the pressures (Reference 15), however the reason for selecting those points were not described. We arbitrarily used the pressure values at the bottom of the “y” descent or the bottom of the “x” descent when the “y” descent could not be identified (Fig 1A and B). This is because measuring ΔPes and ΔCVP at the lowest value of the CVP waveform was the clearest way to distinguish it. However, it is true that there is no data about the optimal CVP trace point to be used for ΔCVP measurement. Selecting other points may improve the accuracy of our estimation method. 

Therefore, in page 7 L115, we replaced the sentence: 

“pressure values at the bottom of the cardiogenic oscillations were used in the calculations (Fig 1).”

by

measurements taken at the bottom of the “y” descent or the bottom of the “x” descent when the “y” descent could not be identified. (Fig 1). 

We also added the following sentence to the limitation in Page 18 L299: 

Thirdly, we selected the bottom of “x” or “y” descent of the cardiogenic oscillations to measure ΔCVP and ΔPes in this study. Selecting other point may improve accuracy of our estimation method [15]. 

EQ-3 Please report the K value also for dPes and discuss which factors influence the K "transmission" factor (chest wall? baseline PCV? PEEP?)

Thank you for pointing this out. We added k and ΔCVP in the occlusion test to Table 2. 

We tried to look for correlations with the information on factors affecting k. We checked the relationships between k and body mass index, CVP, PEEP, and age (Figure), and found no significant relationship between them. As the reviewers pointed out, there is a possibility that respiratory system elastance (Ers) and/or chest wall elastance (Ecw) may affect k; however, we did not measure Ecw. Therefore, we could not draw any conclusions from our data regarding the factors affecting k. 

In the literature, several factors may affect k. Lung volume, chest wall elastance, chest wall distortion, air-trapping, and volume status, including CVP, have been reported to affect relationships between ΔPpl, ΔPes, and ΔCVP (Reference 12,21,26-28). Moreover, the pattern of blood flow into the right atrium may not be the same during airway occlusion and mechanical ventilation (Reference 15), which may affect the pressure and compliance of the right atrium and, as a result, ΔPpl/ΔCVP. Therefore, we added the following sentences to the discussion: 

P16 L248

Lung volume, chest wall elastance, chest wall distortion, and volume status including CVP and air trapping may affect the relationships between ΔPpl, ΔPes, and ΔCVP [12,21,26-28]. 

P18 L307

Moreover, the pattern of blood flow into the right atrium may not be the same during airway occlusion and mechanical ventilation [15], which may affect the pressure and compliance of the right atrium and, as a result, ΔPpl/ΔCVP may also be affected.

EQ-4 Please discuss the impact of chest wall compliance on your results, and if it is related with thoracic/cardiac surgery or different PS levels.

Thank you for pointing this out. As the reviewer noted, the fact that many of the patients in our study were post-operative cardiac surgery infants may have influenced our results. Chest wall elastance is certainly affected by surgery. However, the influence of cardiac surgery may vary (usually elastance increase but may decrease if there is sternal instability). Therefore, we believe that the effect of chest surgery itself on k is unpredictable.

We have added the following sentences to our discussion:

P17 L287

Fourth, since there were many postcardiac surgery patients in our patient group, it is possible that hematomas, adhesions in thoracic cavity, and indwelled pleural catheters (with negative pressure of 5-7 cmH2O) may have influenced the relationship between ΔPao and ΔPes during occlusion. Finally, the large distortion of the thorax in neonates by inspiratory efforts may affect the relationship between ΔPao and ΔPes during occlusion [21]. 

As the reviewers pointed out, PS levels may also affect our results. Therefore, we compared ΔPes and cΔCVP-derived ΔPpl at each PS level (Fig 3). The results show that ΔPes and cΔCVP-derived ΔPpl are always moderately correlated at all PS levels. Theoretically, the PS level may influence these relationships, but our results show the same tendency at all PS levels. 

We added the following sentences to our methods, results and discussion:

P8 L135

The correlation of ΔCVP and cΔCVP-derived ΔPpl with ΔPes at each PS level was also compared. The regression coefficients were also calculated.

P9 L161

The correlation of ΔCVP and cΔCVP-derived ΔPpl with ΔPes in each PS level was tested by Pearson's product-moment correlation coefficient. 

P9 L179

The difference of cΔCVP-derived ΔPpl to ΔPes tended to be smaller than that of ΔCVP to ΔPes in all settings (-0.1 ± 1.5 vs. 3.1± 3.5 cmH2O in PS10, -0.7 ± 3.3 vs. 4.5± 3.9 cmH2O in PS5, and -1.0 ± 3.4 vs 4.7± 4.4 cmH2O in PS0).

P10 L185

The correlation of cΔCVP-derived ΔPpl with ΔPes was not perfect but was slightly stronger than the correlation of ΔCVP with ΔPes at all PS levels (Figure 3).

P18 L303

However, to be precise, this assumption may not always be true. Because PEEP affects lung volume and lung volume affects ΔPpl/ΔCVP [28], ΔPpl/ΔCVP during the occlusion test with no PEEP is different from that during mechanical ventilation with PEEP.

EQ-5 As noted by the reviewers, it is not possible to pool more measurements from the same patients together. Analysis and graphs need to be changed accordingly. Also consider RM Anova analysis to explore differences.

Thank you for the advice. We redid the statistics. Repeated measurement correlation was performed because multiple measurements (Reference 22) were taken on individual patients, as shown in Fig 2, which confirmed our hypothesis that our correction method can estimate ΔPpl by ΔCVP.

Therefore, we rewrote the statistics section on pages 8–9 as follows:

P8 L155

Continuous variables are presented as the mean ± standard deviation. We sought to determine whether there was a linear relationship between cΔCVP-derived ΔPpl with ΔPes. Repeated measurement correlation was performed because multiple measurements were taken on individual patients [22]. To assess the accuracy and precision of predicting ΔPes for ΔCVP and cΔCVP-derived ΔPpl, we performed descriptive statistics on the difference between the ΔPes and the two methods (ΔCVP and ΔCVP-derived ΔPpl). The correlation of ΔCVP and cΔCVP-derived ΔPpl with ΔPes in each PS level was tested by Pearson's product-moment correlation coefficient. Repeated measurement correlation was performed using R (The R Foundation for Statistical Computing, Vienna, Austria). Other statistical analyses were performed using EZR version 1.36 (Saitama Medical Center, Jichi Medical University, Saitama, Japan), which is a graphical user interface for R. A p-value <0.05 was considered statistically significant.

We rewrote the results section on page 10, L183 as follows:

The repeated measures correlation between cΔCVP-derived ΔPpl and ΔPes showed that cΔCVP-derived ΔPpl had good correlation with ΔPes (r = 0.84, p< 0.0001) (Figure 2).

We also replaced the figure legends as follows:

P13 L200

Fig 2. Scatter plots for the repeated measures correlations between cΔCVP-derived ΔPpl and ΔPes. 

For comparison, data from individuals are colored differently. The dots represent data for each patient and corresponding lines represent linear relationship for each patient. 

ΔCVP, change in central venous pressure; ΔPpl, change in pleural pressure; ΔPes, change in esophageal pressure; cΔCVP-derived ΔPpl, ΔPpl calculated using a corrected ΔCVP.

P14 L209

Fig 3. Relationship between cΔCVP-derived ΔPpl or ΔCVP and ΔPes.

A: PS10, B: PS5, C: PS0. In each figure, filled circle represent ΔCVP-derived CVP and open circle represent ΔCVP.

EQ-6. Please ensure that your manuscript meets PLOS ONE's style requirements, including those for file naming. 

We have done as you suggested. Thank you very much.

EQ-7. Please state in your methods section when you conducted this study.

Thank you for pointing this out. We performed the study from March to June 2017 and added this information to the methods.

P5 L86

and had an esophageal balloon catheter placed for clinical purposes between March 2017 and June 2017.  

R1-1. please, the authors should omit to say" proof of the concept" in the title, as this concept is known since several years and dacades.

You made a valid point. Along with the opinions of the three reviewers, the title was changed to the following:

“Estimation of change in pleural pressure in assisted and unassisted spontaneously breathing pediatric patients using fluctuation of central venous pressure: A preliminary study”.

R1-2. the authors should add to table one the values of delta CVP. this would allow to compute the Paw during the occlusion. 

Thank you for pointing this out. We have added ΔCVP values during the occlusion test to Table 2 as you noted. 

R1-3. Please, discuss the possible reasons of the great k variability

Thank you for your comments. The editor also asked a similar question regarding k variability. Therefore, please look at our response to EQ-3.

R2-1. The authors should explain their choice of using an air-filled balloon catheter system instead of a saline or water-filled catheter system, which has been the more common method used in infants. 

Thank you for the comments. As we described in the methods, we used the catheters that were inserted for clinical purposes. This balloon catheter is the only commercially available and approved equipment for measuring esophageal pressure in Japan. In addition, our initial target group was children under 18 years of age; and we believe that the use of the balloon was common in that age group. However, as the reviewer pointed out, it is possible that liquid-filled catheters may be more accurate than balloons; and there may have been less exclusion if our study was performed with a liquid-filled catheter.

We added these comments to the discussion:

P17 L283

Second, we used a balloon catheter instead of a liquid-filled nasogastric catheter, which may be more accurate in small infants [27,30]. Third, the size and volume of the balloon may not have been appropriate for infants [31]. However, this balloon catheter is currently the only commercially available equipment for measuring the Pes of infants in Japan.

R2-2. P. 6: The authors should cite the original literature describing the modern technique for recording esophageal pressure and the occlusion test itself. 

Thank you very much. We agree with you and have incorporated this suggestion into the Methods section of our paper. We replaced the reference by Reference7, 20 (P6 L102).

R2-3. P. 7, first paragraph: It is important to make certain that Paw, Ppl, and CVP are measured at the same end-expiratory volume (which is hard to do in ventilated patients unless flow and volume are accurately recorded). Any increase or decrease in lung volume will over- or underestimate pressure swings. The authors do not describe monitoring flow and volume during their experiments.

First, we would like to confirm that our Paw, Pes, and CVP were simultaneously recorded and measured in the same breath and thus compared under the same end-expiratory lung volume.

However, you are right. PEEP, PS, and the flow at the end of expiration affect lung volume. Changes in lung volume affect k and the relationship between ΔPaw and ΔPes during occlusion and ΔPaw/ΔCVP during mechanical ventilation. Therefore, k during occlusion is not actually the same as ΔPaw/ΔCVP during mechanical ventilation.

To clarify this, we changed the following sentence in P7 L126,

from

After 5 min of stabilization under each ventilator setting, we measured ΔPes and ΔCVP under 10, 5, and 0 cmH2O of pressure support. 

to:

After 5 min of stabilization under each ventilator setting, we measured ΔPes and ΔCVP of the same breath under 10, 5, and 0 cmH2O of PS (Fig 1B).

We also added the following comments to the discussion:

P16 L248

Lung volume, chest wall elastance, chest wall distortion, and volume status including CVP and air trapping may affect the relationships between ΔPpl, ΔPes, and ΔCVP [12,21,26-28]. 

P18 L303

However, to be precise, this assumption may not always be true. Because PEEP affects lung volume and lung volume affects ΔPpl/ΔCVP [28], ΔPpl/ΔCVP during the occlusion test with no PEEP is different from that during mechanical ventilation with PEEP. 

R2-4. P. 7: The authors made measurements at different pressure support levels (line 6) – this will alter end-expiratory volume, resulting in changes in pressure swings as described above. In addition, as the authors point out later, less pressure support should result in greater chest wall distortion because of greater diaphragmatic contraction, leading to

greater discrepancy between Paw and Ppl and CVP changes. Contrary to what the authors assumed, such changes do not produce “a steady state” (line 8).

Thank you for pointing out that k measured at the FRC level is not necessarily the same at the altered end-expiraotry lung volume by changing the PEEP or PS levels. You are correct.

Therefore, we replaced "same" with "similar" in P7, L124, and P7, L131. Also, we removed "stead state" to avoid confusion, as we intended to mean resting breathing. In P7, L124, we wrote “because ΔPaw should be equal to ΔPpl during airway occlusion.”, but replaced it with “because ΔPaw should be equal to, or at least close to ΔPpl during airway occlusion unless there is severe chest wall distortion or air-trapping [18,21]. “

And we also added the following sentences to the limitation in page 18 L301. 

Fourth, we assumed that k (ΔPpl to ΔCVP ratio) obtained during the occlusion test was similar to that obtained during mechanical ventilation. However, to be precise, this assumption may not always be true. Because PEEP affects lung volume and lung volume affects ΔPpl/ΔCVP [28], ΔPpl/ΔCVP during the occlusion test with no PEEP is different from that during mechanical ventilation with PEEP.

R2-5. P. 7: Again, same problem: Authors state that changes in Pes are assumed to be the same as changes in Ppl – not true if there is chest wall distortion (see below) or increased airway resistance. Changes in Pes and secondarily cCVP can only be used as a gross estimate of pleural pressure changes.

As you pointed out, it is possible that the large distortion of the thorax in neonates by inspiratory efforts may have influenced k and the relationship between ΔPes and ΔPpl. Thank you for your suggestions.

We added the following sentences to the discussion:

P16 L248

Lung volume, chest wall elastance, chest wall distortion, and volume status including CVP and air trapping may affect the relationships between ΔPpl, ΔPes, and ΔCVP [12,21,26-28].

P17, L279

First, in infants, because chest wall is more compliant than in adults and inspiratory efforts easily distort chest wall inward direction, it is shown that ΔPes is not necessarily equivalent to ΔPaw (mean pleural pressure swings) during occlusion test in the presence of distortion [21]. 

R2-6. P. 8: Results: Please explain why 6 of 14 patients were excluded. Was it just because of chest wall distortion? Perhaps the occlusion test ratio would have been closer to unity had water-filled catheters been used. Authors should comment on the exclusions.

Thank you for your comments. Please look at our response to the editor (EQ-1)

R2-7. P. 9: Table 1: Six of 8 patients underwent cardiac surgery which can cause distortion of pleural pressure distribution because of pleural effusions and adhesions. Authors should describe if pleural catheters were in place and if they were attached to suction – this would really affect their measurements.

Thank you for your insightful comments. As you pointed out, because there were many post-cardiac surgery patients in our patient group, it is possible that hematomas and adhesions may have influenced the relationship between ΔPpl and ΔPes or ΔCVP. Furthermore, as noted by the reviewer, pleural catheters (with negative pressure of 5–7 cmH2O) were inserted in all postcardiac surgery cases. This may also have an impact in the current study. 

Therefore, we added the following comments to discussion:

P17 L287

Fourth, since there were many post-cardiac surgery patients in our patient group, it is possible that hematomas, adhesions in the thoracic cavity, and indwelled pleural catheters (with negative pressure of 5–7 cmH2O) may have influenced the relationship between ΔPao and ΔPes during occlusion.

R2-8. P. 9: Another point --- if indeed patients had pleural catheters in place, why not directly compare the changes in Ppl with changes in CVP? 

Thank you for your comments. Yes, it is possible to measure pleural pressure directly with a chest drain tube if the drain is filled with liquid. However, our goal is not to measure pleural pressure in postoperative patients. We are considering its application in pediatric ARDS patients. Therefore, we did not use thoracic drains for pressure measurement.

R2-9. Figs. 2 and 3: I counted 22 data points in both plots. How many recordings did the authors obtain in each patient? Are the data points evenly distributed amongst the 8 patients? 

We are very sorry. We forgot to describe the data excluded. In this study, we took data at three points in all cases: PS0, PS5, and PS10. In the previous manuscript, we excluded the PS10 values for the two cases where the respiratory effort was so small that the ΔCVP could not be measured from the CVP waveform. However, in response to the reviewer's suggestion, we decided to include these cases as well, with ΔCVP = 0. Therefore, in the new manuscript, we re-analyzed the data with 24 data points.

R2-10. Again, figures: The authors should explain the outliers in the correlation and Bland-Altman plots. There are at least 4, possibly more outliers. The authors should explain why these data points varied so much from the main group. Possible explanations could include underlying clinical conditions (pneumonia, heart failure, pleural effusions), the respiratory rate, pressure support level and intravascular volume affecting the CVP.

Thank you for pointing out this important fact. We are sorry that we do not have the answer to this question. However, as the reviewer pointed out, our method has several limitations. Most importantly, ΔPes/ΔCVP, k and the relation of these were affected by several factors. Please look at our answers to EQ-3 and EQ-4.

 

R3-1. Can you better define "haemodynamic stable" patient? How were patients with arrhythmia considered? Were they excluded?

Yes, patients with arrhythmia were excluded. Therefore, to clarify this, “hemodynamic stable” was replaced by “with sinus rhythm, were not supported with high-dose catecholamines (more than 0.05 mcg/kg/min of epinephrine equivalent)” (P5 L81)”.

R3-2. In your previous manuscript (https://doi.org/10.1007/s10877-019-00368-y) you stated that "Pressure values at the peak of the cardiogenic oscillations were used for calculation" while in this article you stated that pressure values at the bottom of the cardiogenic oscillations were used in the calculations". Why did you change your method?

Thank you for pointing out this important point. Please look at our response to the editor (EQ-2). Actually, we started the measurements at the peak in our preliminary experience, but it failed because the peak during inspiratory efforts (negative pressure) is difficult to find.

R3-3. In Figure 1A you reported the pressures waveforms recorded during an occlusion test: Δpaw is significantly greater than Δpes (it seems ratio of Δpes to ΔPaw 0,7). Can you explain this?

Sorry, we took the wrong one. The figure was not appropriate. We replaced it. The ratio of ΔPes to ΔPaw in this case was 0.85 (Fig 1A, P8 L144).

R3-4. You calculated k as the ratio of Δpaw to ΔCVP obtained during an occlusion test. Why did you use Δpaw instead of Δpes?

Thank you for your comments. Indeed, if once Pes is measured at some point in spontaneous/assisted breaths, it is possible to use ΔPes/ΔCVP to predict ΔPes from ΔCVP thereafter. It is also true that ΔPao and ΔPes during the friction test are not perfectly consistent with each other. However, the purpose of the present study was to estimate ΔPpl without a Pes catheter (because Pes balloon is not always available). Therefore, we do not intend to use Pes for calibration. 

R3-5. ΔPaw isn’t exactly equal to Δpes and therefore k would be different if Δpes was the numerator instead of ΔPaw. Can you perform calculation with k = Δpes/ ΔCVP?

You are right, ΔPaw during occlusion is not exactly equal to ΔPes. However, as we mentioned above, the purpose of this study is to estimate ΔPpl without using the Pes balloon. Therefore, performing the calculation k = ΔPes/ΔCVP has little meaning to our study. However, we created a graph for the reviewers comparing ΔPes and ΔPpl calculated by k' = ΔPes / ΔCVP. Certainly, R of using Pes for k’ is a little bit better than R when using Pao for k. However, the R values seemed to be very small.

R3-6. Can you add to figure 1B the Paw wave and highlight the change in Paw during breathing?

Thank you for your comments. We have added the Paw waveforms to Figure 1B as you pointed out.

R3-7. k value (range 1.59-3.79) varies a lot between subjects, how do you explain this heterogeneity?

Thank you for the comments. Please look at our comments to the editors (EQ-3).

R3-8. Mean k value is between 2-3 that means ΔPaw is 2-3 times greater than Δpes during an occlusion test. How do you explain this ratio? 

Thank you for the comments. However, we think that the reviewer may have misunderstood this. The ΔPes/ΔPaw during occlusion in our enrolled patients is always between 0.8 and 1.2, as we stated in the discussion and figure 1. We think that ΔCVP dissociating from ΔPaw (or ΔPpl) during occlusion indicates that ΔPpl is not directly reflected in ΔCVP. In previous papers, large dissociation of ΔPes and ΔCVP (Reference 14, 17) and various ratios of ΔPaw / ΔCVP during the occlusion test (Reference 13) have been reported.

R3-9. It could be that the pressure drop in the chest increases the venous return, with the heart not handling it and the PVC rising (so that the PVC delta is relatively low). Interestingly, the only one non-cardiopatic patient has the lowest k value: please comment.

Thank you for your insightful comments. As noted by the reviewer, spontaneous breathing may cause more blood flow into the right atrium during airway occlusion than otherwise. This may lead to dilatation and stiffening of the right atrium, which in turn may make it more difficult for pressure in the thoracic cavity to be transferred into the right atrium and into the vessels. Therefore, ΔPpl/ΔCVP during airway occlusion may be different from ΔPes/ΔCVP during normal breathing. 

This was also added to the limitation as follows (P18, L307):

Moreover, the pattern of blood flow into the right atrium may not be the same during airway occlusion and mechanical ventilation [15], which may affect the pressure and compliance of the right atrium and, as a result, ΔPpl/ΔCVP may also be affected.

R3-10. You excluded more than 42% of scrreened children because esophageal balloon position could not be confirmed during the occlusion test. It’s a very high rate, what’s your explanation? Didn’t position and inflation adjustment resolve the discordance?

Thank you for the comments. Please look at our response to the editor (EQ-1).

R3-11-1. Babies who receive pressure support are in assisted spontaneous breathing patients. You obtained 16 measurements during assisted ventilation and only 8 during unassisted spontaneous breathing. Please make it explicit in the title and text

Your point is well taken, so I changed the title to " Estimation of change in pleural pressure in assisted and unassisted spontaneous breathing pediatric patients using fluctuation of central venous pressure: A preliminary study. " and changed several parts of the abstract and text.

R3-11-2. You analyzed all obtained measurements together: it can be methodologically incorrect. You have to analyze measurements obtained in different settings separately because pressure support magnitude can differently influence Δpes and ΔCVP. Please provide three different analysis: pressure support 10, 5 and 0.

Thank you for your important comments. As you pointed out, our method is not correct. Therefore, we analyzed the relationship between ΔPes and cΔCVP-derived ΔPpl with each PS setting. the results are shown in Figure 3. We have corrected all relevant parts of this statistical error.

R3-11-3. During data collection did you change only pressure support or did you modify other ventilator parameters 

This is also an important point. The same conditions were used for all ventilator settings except for PS. We added the following sentence to P7, L128, 

The other ventilator settings were unchanged during the measurements. 

R3-11-4. Can you add a table to illustrate how other parameters (respiratory rate, paO2, paO2/FiO2, paCO2, pH, heart rate, blood pressure, capillary refill, sedation level) modified during pressure support modification?

Thank you for the comments. The available parameters that you pointed out were added to Table 1. The sedation level was SBS-1 when all data were measured and added to the methods on page 7 L117. 

All measurements were performed with level -1 sedation on the State Behavioral Scale.

R3-12. For each patient add ventilator parameters (Pplateau, mean airway pressure, PEEP, tidal volume, respiratory rate, minute volume), Δpes, Δpaw, ΔCVP, cΔCVP-derived Δppl, ΔPes-derived plateau PL cΔCVP-derived plateau PL recorded at the moment of data collection.

We are sorry that we do not have some of the data that you pointed out. All available data are listed in Tables 1 and 2.

R3-13. Sample size: the small number of patients enrolled reduces the attendibility of statistical analysis. Bland-Altmann is not validated for analysis of so few repeated measures. Discuss this item. Furthermore agreement assessed with the Bland-Altman analysis should be described as the median difference (bias) and 2.5th and 97.5th percentiles (95%-limits of agreement [LoA] (Bland JM, Altman DG. Statistical methods for assessing agreement between two methods of clinical measurement. Lancet 1986;1: 307-310)

You are totally right. Bland-Altmann is not validated for the analysis of very few repeated measures. In response to various statistics, we reanalyzed the analysis as follows.

First, a comparison between our method and the direct ΔCVP method was added to the analysis to show that our method is more accurate and precise than the direct ΔCVP method.

Second, as noted by the reviewers, we used a repeated measurement correlation to compare ΔPes and ΔCVP-derived ΔPpl because we collected data from one patient three times, each under different conditions.

Third, to assess the accuracy and precision of predicting ΔPes for ΔCVP and cΔCVP-derived ΔPpl, as noted by the reviewers, we abandoned the BA analysis and performed descriptive statistics on the difference between the ΔPes and the two methods (ΔCVP and ΔCVP-derived ΔPpl).

Lastly, correlation analysis between ΔCVP and cΔCVP-derived ΔPpl with ΔPes was performed for each PS.

According to the above policy, we extensively rewrote the section of “statistical analysis” and described the results according to a new analysis. Most importantly, we added the following sentence to our introduction: 

P5 L69

The aim of this study was to test whether our correction method could improve accuracy to estimate ΔPpl than using raw ΔCVP value and could be used in pediatric patients who have spontaneous breaths during mechanical ventilation.

R3-14. Several studies (Colombo J et al. Detection of strong inspiratory efforts from the analysis of central venous pressure swings: a preliminary clinical study, Minerva Anestesiologica, in press 10.23736/S0375-9393.20.14323-2; Hedstrand U, Jakobson S, Nylund U, Sterner H. The concordance of respiratory fluctuations in oesophageal and central venous pressures. Ups J Med Sci 1976;81: 49-53; Flemale A, Gillard C, Dierckx JP. Comparison of central venous, oesophageal and mouth occlusion pressure with water-filled catheters for estimating pleural pressure changes in healthy adults. Eur Respir J 1988;1: 51-57; Chieveley-Williams S, Dinner L, Puddicombe A, Field D, Lovell AT, Goldstone JC. Central venous and bladder pressure reflect transdiaphragmatic pressure during pressure support ventilation. Chest 2002;121:533-538; Bellemare P, Goldberg P, Magder SA. Variations in pulmonary artery occlusion pressure to estimate changes in pleural pressure. Intensive Care Med 2007;33:2004-2008; Verscheure S, Massion PB, Gottfried S, Goldberg P, Samy L, Damas P, et al. Measurement of pleural pressure swings with a fluid-filled esophageal catheter vs. pulmonary artery occlusion pressure. J Crit Care 2017;37:65-71) report a poor agreement but a positive and significant correlation between ΔCVP and ΔPES: even if ΔCVP and ΔPES were not always the same, smaller or larger ΔCVP generally reflected smaller or larger ΔPES. Starting from your data and the manuscripts above you shold better discuss this theme and explain what your work add to previous data

In small children chest x-ray cannot confirm the exact position of CVC tips immediately above the SVC-right atrium junction. This could produce a bias in CVP measurements. 

Thank you for your kind suggestion. We included those previous studies to our introduction on page 4 L60 as follows.

As a potential surrogate for the detecting the change in Ppl (ΔPpl) or strong inspiratory efforts, the change in central venous pressure (ΔCVP) has been repeatedly examined [10–18]. However, inconsistent results in previous papers have shown ΔCVP to be both an underestimation and an overestimation of ΔPpl [10, 12, 14–18]. Accordingly, ΔCVP has not been generally accepted as a surrogate for ΔPpl. 

R3-15. I think electrocardiogram (ECG)-guided technique could be a more precise technique. Why didn't you used this approach?

Thank you for the comments. It is true that the ECG method can be used to insert the device into a more reliable position. However, in this study, we used a CV line that was already inserted, so we only confirmed it with Xp. 

R3-16. Are all patients in respiratory weaning or some of them are in an acute phase of respiratory illness? Pleas add this information in population description

Thank you for your question. To clarify your question and describe our patients in detail, we added the reason for intubation, length of mechanical ventilation, days from intubation to study enrollment, and length of ICU stay to Table 1.

R3-17. Most patients are post-cardiac surgery ones: was diaphragmatic function normal in all of them?

We are sorry that we did not perform ultrasound evaluation. However, all cases were successfully extubated, and there were no clinically diagnosed cases of phrenic nerve palsy.

R3-18. Please add the correlation test you used to performed analysis

We used Pearson's product-moment correlation for the correlation test and added this to our methods on page 9 L161:

The correlation of ΔCVP and cΔCVP-derived ΔPpl with ΔPes in each PS level was tested by Pearson's product-moment correlation coefficient. 

R3-19. In table 1 add patients’ age, reason to PICU admission, initial severity of disease, days from icu admission to study enrollment

Add table to describe main characteristics of the study population at study entry: FiO2, Arterial pH, Arterial CO2 tension, Arterial O2 tension, Arterial O2 saturation, PaO2:FiO2, Heart rate, Mean arterial pressure, Central venous pressure, Central venous O2 saturation, Lactate, Urinary output, Vasopressors, Clinical evaluation of volemic status (Hypovolemic, Normovolemic, Hypervolemic)

We apologize that some of the data you requested could not be obtained. However, we added some information to Tables 1 and 2.

R3-20. The work is of some interest but it has some methodological limitations (also related to the statistical analysis) that needs to be addressed. I think, due to the small sample size, the most appropriate article type for the manuscript is a preliminary report.

Thank you for the comments. According to your suggestion, we changed the title.

We also redid the statistics; then, the abstract, method, and results were rewritten. Please check the new abstract, statistical analysis section, and results.

 

R4-1. The use of this correction factor with which the authors derive CVP-derived Pl is imaginative and perhaps is responsible for the bias they found. It would be very helpful to report the ratios they also found in the fall in CVP to the fall in pleural pressures as measured by the fall in Peso. They mention that they should be the same as the ratio between CVP and airway but were they?

As the reviewer pointed out, our measured k and ΔPes/ΔCVP are not exactly the same. The reviewer is absolutely correct. Therefore, we replaced "same" with "similar" in P7, L124, and 130. However, although there are several limitations to our hypothesis, we believe that correction by k can improve the accuracy of estimating ΔPpl from ΔCVP, as shown in Figure 3. Therefore, we have added the following sentences on page 18 L301:

Fourth, we assumed that k (ΔPpl to ΔCVP ratio) obtained during the occlusion test was similar to that obtained during mechanical ventilation. However, to be precise, this assumption may not always be true. Because PEEP affects lung volume and lung volume affects ΔPpl/ΔCVP [28], ΔPpl/ΔCVP during the occlusion test with no PEEP is different from that during mechanical ventilation with PEEP. Moreover, the pattern of blood flow into the right atrium may not be the same during airway occlusion and mechanical ventilation [15], which may affect the pressure and compliance of the right atrium and, as a result, ΔPpl/ΔCVP may also be affected. However, the ratio of ΔPes/ΔCVP to k during occlusion at PS10, PS5, and PS0 were acceptable (0.86±0.31, 0.98±0.27, and 0.95±0.34, respectively) in this study. 

R4-2. Furthermore, the authors recognize the limitations of the study and its generalizability given the few subjects and are cognizant of the possible limitations of using the fall in CVP at higher levels. Whereas their "indexation" should work at more modest CVP values, during the airway occlusion manueuver the diaphragms should not move caudal and propel blood from a charged splanchnic circulation into the right atrium which, during spontaneous ventilation, could further dilate the right atrium, increase its pressure, and make transmission of pleural pressures to the non-compliant right atrium even more problematic.

 Thank you for the great suggestion. As noted by the reviewer, spontaneous breathing may cause more blood flow into the right atrium during airway obstruction than otherwise. This may lead to dilatation and stiffening of the right atrium, which in turn may make it more difficult for pressure in the thoracic cavity to be transferred into the right atrium and into the vessels. Therefore, ΔPpl/ΔCVP during airway occlusion may be different from ΔPes/ΔCVP during normal breathing. 

This was also added to the limitations as follows:

P18, L307

Moreover, the pattern of blood flow into the right atrium may not be the same during airway occlusion and mechanical ventilation [15], which may affect the pressure and compliance of the right atrium and, as a result, ΔPpl/ΔCVP may also be affected. 

R4-3. Finally, I am curious on how the authors measured both the CVP - to which landmark was the vascular transducer leveled - and how the fall in CVP was measured. In terms of the latter, it appears from the graph 1A, that the authors use the bottom of the "Y descent" as the land mark and I'm not sure that is appropriate; using the base of the "a" wave or, failing that, the mean CVP would appear superior.

Thank you for your comments. First, we added the following sentence to “setting for measurement and recording”

The pressure transducer for CVP measurement was leveled at mid-axillar line. (page 6, L91)

Regarding the landmark, please look at our comments to the editor (EQ-2).

---

## [Decision Letter · Decision Letter 1]

8 Feb 2021

Estimation of change in pleural pressure in assisted and unassisted spontaneous breathing pediatric patients using fluctuation of central venous pressure: A preliminary study.

PONE-D-20-20085R1

Dear Dr. Takeuchi,

We’re pleased to inform you that your manuscript has been judged scientifically suitable for publication and will be formally accepted for publication once it meets all outstanding technical requirements.

Please add the missing reference suggested by rev n.2

Kind regards,

Andrea Coppadoro

Academic Editor

PLOS ONE

Reviewers' comments:

Reviewer's Responses to Questions

**Comments to the Author**

1. If the authors have adequately addressed your comments raised in a previous round of review and you feel that this manuscript is now acceptable for publication, you may indicate that here to bypass the “Comments to the Author” section, enter your conflict of interest statement in the “Confidential to Editor” section, and submit your "Accept" recommendation.

Reviewer #2: (No Response)

Reviewer #3: All comments have been addressed

Reviewer #4: All comments have been addressed

2. Is the manuscript technically sound, and do the data support the conclusions?

Reviewer #2: Yes

Reviewer #3: Yes

Reviewer #4: Yes

3. Has the statistical analysis been performed appropriately and rigorously? 

Reviewer #2: Yes

Reviewer #3: I Don't Know

Reviewer #4: I Don't Know

4. Have the authors made all data underlying the findings in their manuscript fully available?

Reviewer #2: Yes

Reviewer #3: Yes

Reviewer #4: Yes

5. Is the manuscript presented in an intelligible fashion and written in standard English?

Reviewer #2: Yes

Reviewer #3: Yes

Reviewer #4: Yes

6. Review Comments to the Author

Reviewer #2: The authors have satisfactorily and convincingly responded to reviewers' comments. They are to be commended for conducting a technically difficult but sophisticated study which can also be applied to older children and adults.

The only thing I would add is an additional, very relevant reference by Baydur et al, Validation of the esophageal balloon technique at different lung volumes and postures. J Appl Physiol 1987: 62(1):315-321. This study looked at delta Pes/delta Pm in spontaneously breathing adults under different conditions. It is directly related to the topic of this paper.

7. PLOS authors have the option to publish the peer review history of their article (what does this mean?). If published, this will include your full peer review and any attached files.

Reviewer #2: No

Reviewer #3: No

Reviewer #4: No

---

## [Editor Report · Acceptance letter]

17 Feb 2021

PONE-D-20-20085R1 

Estimation of change in pleural pressure in assisted and unassisted spontaneous breathing pediatric patients using fluctuation of central venous pressure: A preliminary study.  

Dear Dr. Takeuchi:

I'm pleased to inform you that your manuscript has been deemed suitable for publication in PLOS ONE. Congratulations! Your manuscript is now with our production department. 

Kind regards, 

on behalf of

Dr. Andrea Coppadoro 

Academic Editor

PLOS ONE